# Metamorphism of Venus as driver of crustal thickness and recycling

Julia Semprich [1] ✉, Justin Filiberto [2], Matthew Weller[3,5], Jennifer Gorce [3,4] & Nolan Clark[3]

The composition and thickness of the venusian crust and their dependence on thermal gradients and geodynamic setting are not well constrained. Here, we use metamorphic phase transitions and the onset of melting to determine the maximum crustal thickness of basaltic plains in different tectonic settings. Crustal thickness is limited to ~40 km in a stagnant lid regime with a low thermal gradient of 5 °C/km due to density overturn and delamination. In contrast, the maximum crustal thickness in a mobile lid regime with a high thermal gradient of 25 °C/km is restricted to ~20 km due to the onset of crustal melting. The thickest the crust can be is ~65 km for a basaltic crust with a thermal gradient of 10 °C/km. Our models show that a venusian basaltic crust cannot be thicker than 20–65 km without either causing delamination and crustal recycling or melting and producing volcanic eruptions.

Venus is unique in our Solar System; its planetary surface is cloaked by omnipresent clouds and a thick ~92 bar atmosphere, which leads to current day surface temperatures exceeding 460 °C[1]. Observations of Venus' surface reveal voluminous volcanic plains, thought to be emplaced within the last billion years[2,3], with ongoing volcanic activity (inferred from observed surface changes[4], infrared, microwave, and thermal emission data[5,6], and weathering rates[7]).

Since surface rocks on Venus are exposed to conditions comparable to contact metamorphism on Earth, the deeper venusian crust is expected to experience phase transitions forming high-grade metamorphic assemblages until partial melting is reached. Metamorphism of the crust significantly changes the geophysical properties of rocks including density and porosity[8,9], which in turn has major implications for crustal density, thickness, geodynamics, crustal recycling mechanisms, and the global tectonic state of Venus[10–12].

Metamorphic reactions are strongly dependent on the composition of the crust, including its volatile content, and thermal gradients. Direct bulk rock measurements (with substantial error bars) from the Venera and Vega landers indicate that both tholeiitic and alkalic basalt may be present on the surface of Venus e.g., refs. [13–15], and emissivity and radar measurements suggest that the bulk surface of Venus is

basaltic in nature[16–18]. The volatile content in the venusian interior is unknown due to a lack of direct measurements. The high deuterium to hydrogen ratio[19] suggests that early Venus had water levels comparable to those on Earth[20] but has lost a significant amount of water after its formation; although, there is substantial uncertainty about the timing and mechanism of this process[21–27]. It is likely that the interior of Venus has retained a small fraction of water and carbon dioxide (and potentially other volatiles), although the initial volatile content will have been depleted due to melting and degassing[10] and may be relatively small (ppm range). Furthermore, studies on fault strength and elastic thickness[28,29] suggest that the present-day venusian lithosphere is predominantly dry. Estimates of the global average heat flow[30–33] are in the range of 10–30 mW/m², corresponding to thermal gradients of ~5–15 °C/km although regions of higher heat flow (>50 mW/m²) are possible, as has been suggested for coronae, rifts, steep-sided domes, and crustal plateaus[11,12,34–36].

The thickness of the venusian crust is also not well constrained and estimates for Venus' average crustal thickness, based on topography and gravity models, range from 8 to 25 km[36–38] with a possible maximum thickness of ~90[39]. Crustal estimates derived from gravity models are particularly sensitive to crustal density[40]. Both

[1]AstrobiologyOU, School of Environment, Earth, and Ecosystem Science, The Open University, Milton Keynes, UK. [2]Astromaterials Research and Exploration Sciences, NASA, Johnson Space Center, Houston, TX, USA. [3]Lunar and Planetary Institute, USRA, Houston, TX, USA. [4]Amentum at JSC, Houston, TX, USA. [5]Present address: Department of Earth and Environmental Sciences, Rensselaer Polytechnic Institute, Johnson Rowland Science Center, Troy, NY, USA. ✉ e-mail: julia.semprich@open.ac.uk

phase transitions and partial melting have significant influence on the thickness of the crust. For example, the transition of basalt to granulite or eclogite mineral assemblages causes a significant increase in density, which limits the topography supported by isostasy[41,42]. When the crustal root becomes denser than the surrounding mantle, it triggers lithospheric delamination[43–46] which is the detachment of the lowermost crust. This process has been suggested as a driving force for gravitational instabilities and magmatism at the crust-mantle boundary in the venusian interior[10,47,48], particularly for coronae[49,50], and tesserae[51]. However, phase equilibria modelling for crustal compositions on Earth has demonstrated that the crustal density significantly varies with pressure, temperature (i.e., the thermal gradient, which is interdependent with the tectonic state, the lithospheric thickness, and heat flux) and composition[52]. Delamination may therefore not always occur at the phase boundary and the assumption of a linear increase in density from gabbro to eclogite e.g., ref. [41] is an inadequate approach. A detailed study of the mineralogical, and associated density changes, expected in the venusian crust is therefore critical to constrain potential crustal thicknesses, as well as crustal recycling via delamination and/or melting.

In this study, we use a phase equilibria modeling approach to calculate the effects of pressure and temperature on the mineralogy for dry basalt and alkali basalt compositions, along with a peridotite mantle composition, to determine the conditions required for melting of the crust and associated density changes of the crust and mantle. Phase equilibria calculations have the advantage in that stable mineral assemblages and their corresponding rock densities reflect gradual changes with depth into the planetary interior, instead of an average, because they are calculated as a function of pressure, temperature, and composition. We then compare the calculated crust and mantle densities to investigate the conditions for potential density inversions (where the crust becomes denser than the underlying mantle). Finally, we compare our results with estimates of crustal thicknesses to predict where either crustal melting or crustal recycling through delamination is likely to occur on Venus. According to out petrological constraints, the basaltic venusian crust cannot be thicker than 20–65 km and is strongly dependent on thermal gradient. For high thermal gradients, the crustal thickness is limited to ~20 km due to the onset of melting which represents are more mobile tectonic regime, while a low thermal gradient of 5 °C/km can cause density overturn and delamination of the crust reflecting a more stagnant-lid tectonic regime.

## Results

### Maximum crustal thickness

Results presented here focus on mafic crust representing venusian volcanic plains in stable conditions excluding specific regions such as tesserae as well as coronae and rifts with significantly higher thermal gradients.

We find that the maximum crustal thickness is strongly dependent on the thermal gradient and ranges from ~21 to 65 km, where the thinnest crust is a result of high thermal gradients of 25 °C/km and the thickest crust is supported by a 10 °C/km thermal gradient (Fig. 1 and Table 1). The 5 °C/km is the only thermal gradient where density overturn and delamination limit the crustal thickness for both basalt and alkali basalt crustal compositions at ~40 km (Fig. 1A) since the crustal density of both exceed that of the peridotite (Fig. 2A). This is mainly a result of the transition of the dry basalt and alkali basalt compositions from an assemblage dominated by plagioclase and pyroxene (Figs. 3B, C and 4B, C) to garnet-bearing assemblages (basalt: clinopyroxene + plagioclase + garnet ± orthopyroxene ± quartz; Figs. 3B and 4B; alkali basalt: clinopyroxene + plagioclase + garnet + kalsilite + spinel + ilmenite; Figs. 3C and 4C). Particularly the formation of dense garnet and the gradual replacement of plagioclase with clinopyroxene in the basalt and alkali basalt result in significant densification with depth compared to the peridotite, where the

mineralogy is dominated by olivine, orthopyroxene, clinopyroxene at most pressure-temperature conditions (Figs. 3A and 4A). The density overturn occurs before either of the crustal compositions transition to a plagioclase-free mineral assemblage. This metamorphic progression is due to the system being completely dry, which will suppress the formation of amphiboles. The full transition to an assemblage dominated by pyroxene and garnet in our models occurs at depths >45 km for the alkali basalt and >53 km for the basalt, which can result in further densification (>3350 kg/m³) of the already delaminating crust. This process could contribute to the heterogeneity of the venusian mantle, but we find it has no impact on the thickness of the crust.

For all other thermal gradients (10–25 °C/km, Figs. 1B–E and 2B–E), the crustal compositions reach the solidus and start melting before reaching the point of density overturn, resulting in a thinner crust with increasing thermal gradient (Table 1). Melt proportions are relatively small close to the solidus of the basalt, but increase rapidly with increasing temperatures (Fig. 4E, H, K, N). The alkali basalt forms significant amounts of melt (26–37 vol%) shortly after melting temperatures are reached (Fig. 4F, I, L, O).

### Density of the crust and mantle

Densities were calculated for the solid rock only, thus excluding melt once the compositions reached the solidus. Our models yield crustal densities of >2900 kg/m³ for near surface conditions with the basaltic composition closer to 3000 kg/m³ (Fig. 2). The average density of the upper crust is relatively constant in all our scenarios, even though the absolute density of the crust can vary with crustal thickness, the thermal gradient, and the composition. This can be exemplified by calculating the average density of a basaltic composition for a crustal thickness of 20 km on the 5 °C/km and 25 °C/km thermal gradients: 2993 kg/m³ (5 °C/km) versus 2963 kg/m³ (25 °C/km), and therefore a difference of 30 kg/m³ or ~10% variation in density. For a crustal thickness of 40 km, the average density for the low thermal gradient is 3085 kg/m³. In contrast, there would have been significant melt formation on the 25 °C/km thermal gradient and the average density of the depleted crust would be ~2975 kg/m³. It should be noted that our petrologically constrained densities are significantly higher than the average density values (2800 and 2900 kg/m³) used in some geophysical and geodynamic models[37–39,53] while more recent studies using equations-of-state to calculate crustal densities[50,54,55] in the range of 2900–3000 kg/m³ are a better match to our results.

The density of the peridotite to a depth of 80 km (upper mantle) is in the range of 3200–3300 kg/m³ for thermal gradients of 10–25 °C/km and only exceeds 3300 kg/m³ by ~40 kg/m³ for the 5 °C/km thermal gradient (Fig. 2). As a result, the density contrast between the crust and mantle is usually in the range of 200–300 kg/m³. However, the density contrast is lower for the 5 °C/km thermal gradient, where the density of the crust increases until it reaches mantle density, and on the 10 °C/km thermal gradient, where the crustal density is comparable to that of the mantle shortly before the onset of melting. Again, these values deviate significantly from the values used in geophysical and geodynamic models (200–300 kg/m³ vs 400–500 kg/m³) used to derive crustal thickness from the relationship between global topography and gravity data[37–39].

While the present-day crustal thickness is limited by the onset of melting, earlier stages in Venus' geological evolution may have recorded melting episodes with implications for crustal compositions, and we have therefore also reported the density of the residual rock assuming that melt is removed from the system (Fig. 2B–E). The restite of the alkali basalt reaches the point of density overturn for thermal gradients of 10 and 15 °C/km at shallower depths than the basalt due to the significant amount of melt formation and subsequent fractionation (Fig. 2B, C). For the 20 and 25 °C/km thermal gradients, the density of the basalt does not exceed that of the peridotite (Fig. 2D, E) and crustal delamination is not feasible even after significant melt extraction.

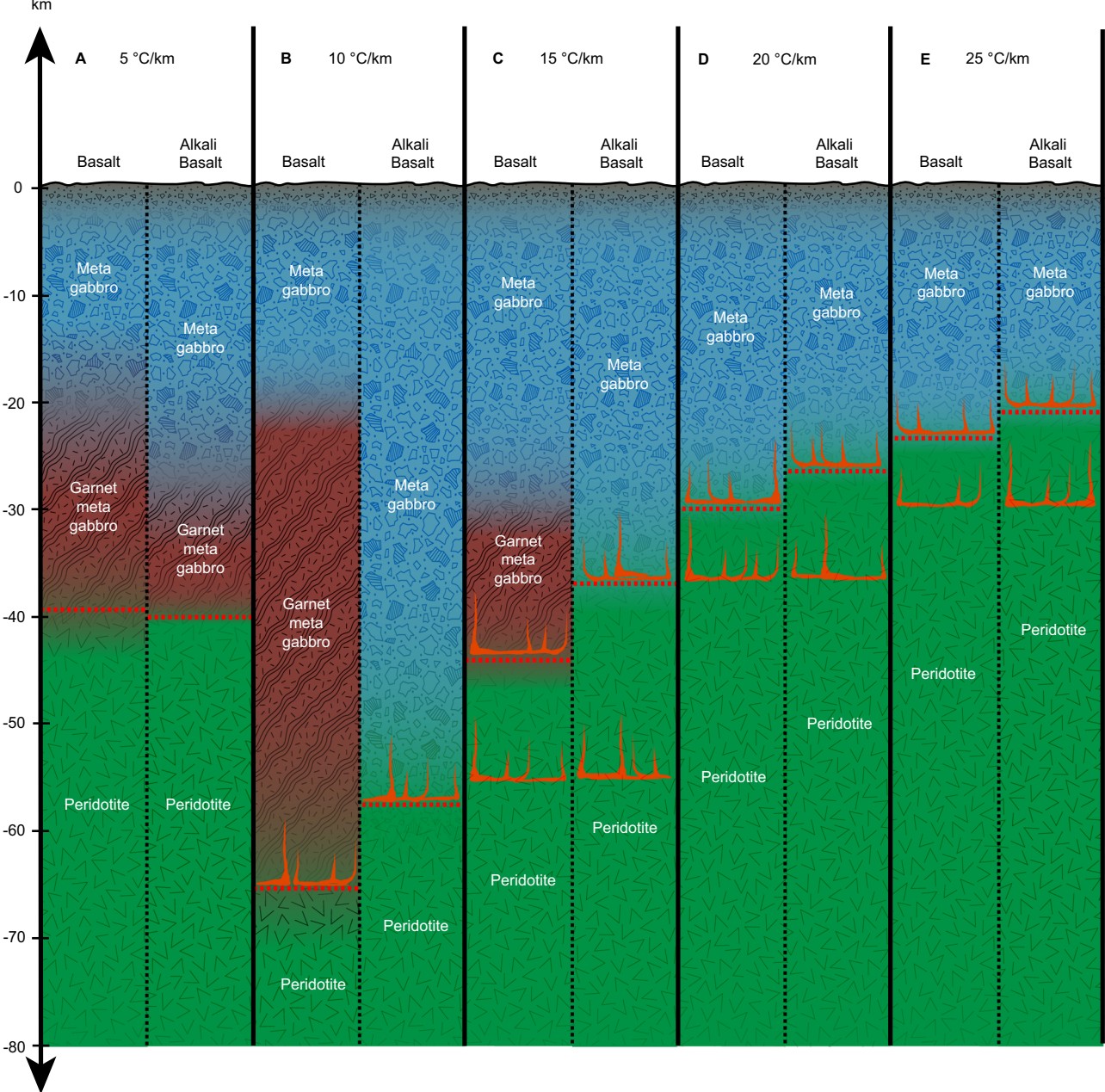

**Fig. 1 | Maximum crustal thickness of venusian crust.** Crustal composition and thickness determined by petrological modeling for basalt and alkali basalt compositions and thermal gradients of 5–25 °C/km. For simplicity, mineral assemblages with garnet are classified as garnet metagabbro (shown in dark red) and assemblages without garnet as metagabbro (shown in blue) while the actual mineralogy may vary. The yellow dotted line indicates the maximum crustal thickness, either defined by a density overturn for the 5 °C/km thermal gradient (**A**) or melting for 10–25 °C/km thermal gradients (**B**–**E**). The onset of melting of the crust and mantle is indicated by ascending magma (red).

## Table 1 | Estimates for maximum crustal thickness determined for basaltic and alkali basaltic compositions and a range of potential venusian thermal gradients

| Thermal gradient (°C/km) | Basalt | Alkali Basalt |
|---|---|---|
| 5 | 39 ± 4 km | 40 ± 4 km |
| 10 | 65 ± 7 km | 57 ± 6 km |
| 15 | 44 ± 4 km | 37 ± 4 km |
| 20 | 30 ± 3 km | 27 ± 3 km |
| 25 | 23 ± 2 km | 21 ± 2 km |

## Discussion

We show that only basaltic compositions with a thermal gradient of ~10 °C/km allows for thickened crust (<65 km). Either delamination processes (<~10 °C/km) or melting (>~10 °C/km) limit maximum crustal thicknesses to <44 km. Crustal thickness estimates of >65 km for basaltic crust at or near planetary radius are therefore not feasible under petrological constraints. These limits may differ for regions with higher elevation (e.g., volcanic rises, crustal plateaus), where isostasy also plays an important role, which was not considered in our models. Many estimates using geoid-to-topography ratios are within this range including average crustal thickness values of ~10–20 km[39], ~10–25 km[37], ~8–25 km[38], and <25 km[56]. However, regional maxima up to 80 km have

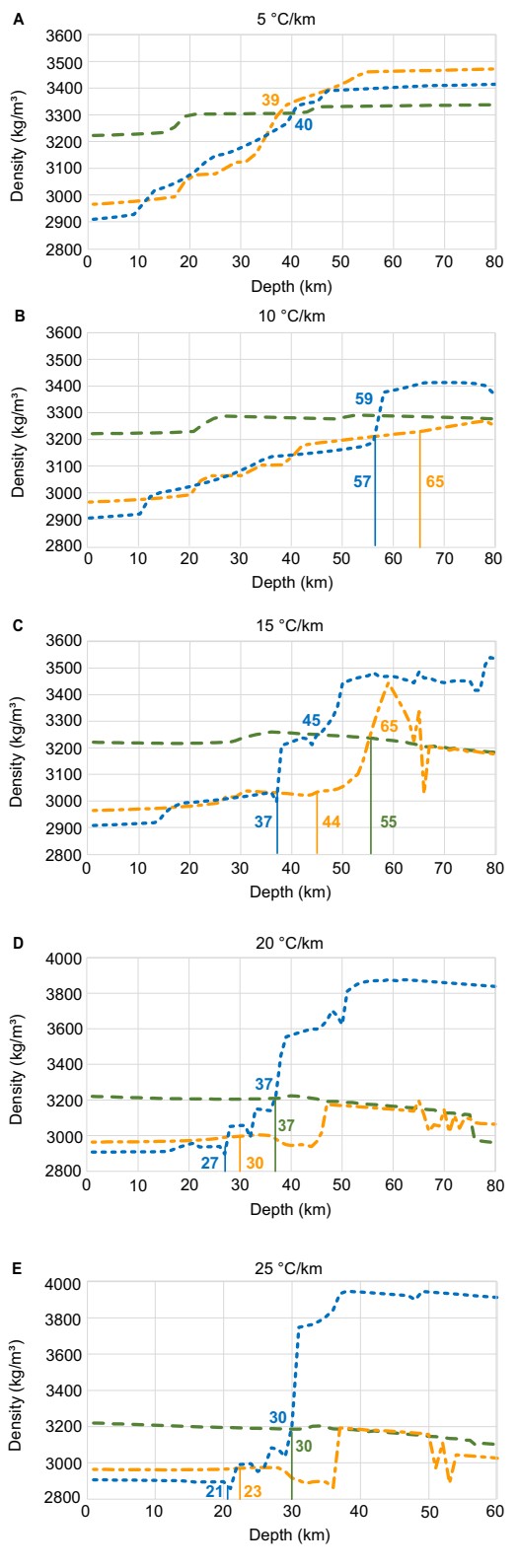

**Fig. 2 | Density variation with depth and thermal gradient.** Comparison of densities for the compositions of peridotite, basalt, and alkali basalt along thermal gradients of 5 °C/km (**A**), 10 °C/km (**B**), 15 °C/km (**C**), 20 °C/km (**D**) and 25 °C/km (**E**). Vertical lines (**B**–**E**) indicate the depth at which melting occurs for each of the compositions (melting in **A** starts at greater depth than shown). Numbers represent the depth (km) at which either density crossover or melting occurs. After the start of melting, densities are reported only for the restite, with the melt being removed from the system. The density scale (y-axis) varies for different thermal gradients.

been reported for the venusian plains (although uncertainty is increased due to low resolution) and some convection models yield estimates in the range of ~60 km for average crust[57], which would restrict the thermal gradient to ~10 °C/km. Crustal thicknesses exceeding the maximum of ~65 km may therefore indicate the presence of more silica-rich compositions, porous materials, or less dense alteration phases within the crust. Silica-rich rocks are only expected to be stable for lower thermal gradients (~5–10 °C/km) due to the lower solidus of these compositions e.g., ref. 58, which would in turn result in re-melting of the crust. The presence of more felsic compositions has been suggested for tesserae[17] but is less likely for basaltic plains, the focus of this study.

While porous rock material has been reported on the surface at some Venera landing sites e.g., ref. 59, there are currently no constraints on subsurface porosity although metamorphic pressure and temperature conditions should enhance compaction in the crust. Alteration phases that could lower the rock density such as sulfates or hydrous phases have been reported in experiments and thermodynamic models at surface conditions e.g., ref. 60, but it is not known if these phases are present in the deeper crust. Further, the present-day venusian lithosphere is assumed to be predominantly dry[28,29], and we have adopted this assumption in our model approach although the influence of volatiles on crustal densities and consequently the crustal thickness should be addressed in future work.

A thermal gradient in the range of 5 °C/km results in a maximum crustal thickness of ~40 km (Table 1 and Figs. 1A and 2A) for basalt and alkali basalt compositions due to density overturn and subsequent delamination of the crust. This densification is driven by phase transitions due to increasing pressure and temperature, but the density overturn occurs even before the crust completely transitions to eclogite in our dry and equilibrated model system. Contrary to previous work, the eclogite transition is therefore not a good proxy to define the maximum thickness of the crust. A low thermal gradient and thick (~40 km) crust will also result in a much lower density contrast (200–300 kg/m³) at the crust-mantle boundary than generally assumed in crustal thickness modelling (400–500 kg/m³)[37–39], which significantly impacts estimates of the average crustal thickness and the derived minimum and maximum crustal thickness values[37].

According to our petrological models, crust thicker than ~40 km in very low thermal regimes (5 °C/km) will be denser than the mantle, which is expected to trigger delamination, the detachment of the lowermost crust and subsequent sinking into the mantle. Numerical and laboratory models using terrestrial parameters have shown that the styles and time scales of delamination depend on various factors including the temperature at the crust-mantle boundary, the viscosity of the crust and mantle, and the density contrast[61–64]. The result of delamination will be the thinning of the lithosphere and upwelling of hot material[43,46], with significant implications on observed heat flow and crustal depth. Further, the delaminated crustal material may contribute to heterogeneities in the mantle. According to our model, partial melting of basalt and alkali basalt would be expected at a depth of 121–197 km and the resulting eclogite melts are geochemically different from peridotite melts[10]. Identifying their geochemical signatures (e.g., generally higher $SiO_2$ and lower MgO) by emissivity measurements of the surface could imply that delamination and crustal recycling occurred at some stage in Venus' geological history.

For thermal gradients of 10 °C/km and 15 °C/km the basaltic and alkali basaltic compositions start melting before their densities reach that of the mantle, which still results in a thick lithosphere of <65 (10 °C/km) and <44 (15 °C/km), respectively. Delamination would therefore not be expected for these thermal regimes, unless there is significant melting and melt fractionation, which results in a dense restitic assemblage that eventually exceeds mantle densities (Fig. 2B, C).

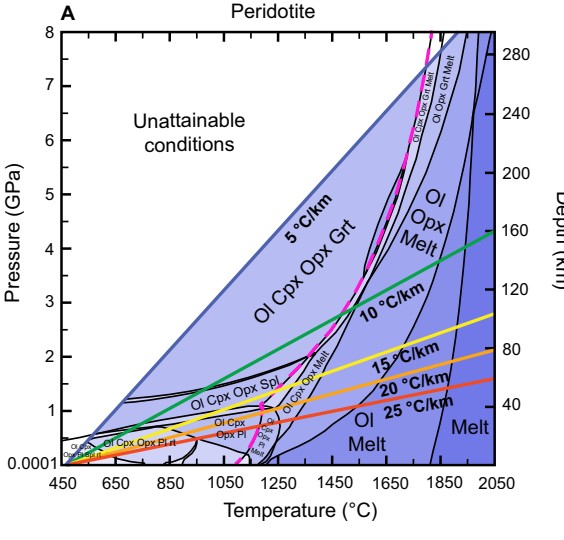

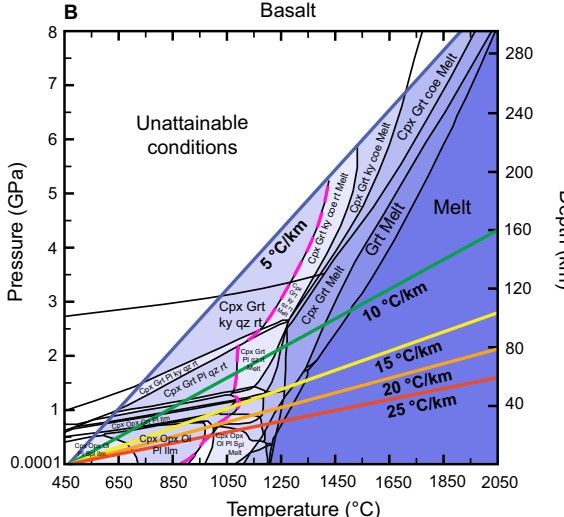

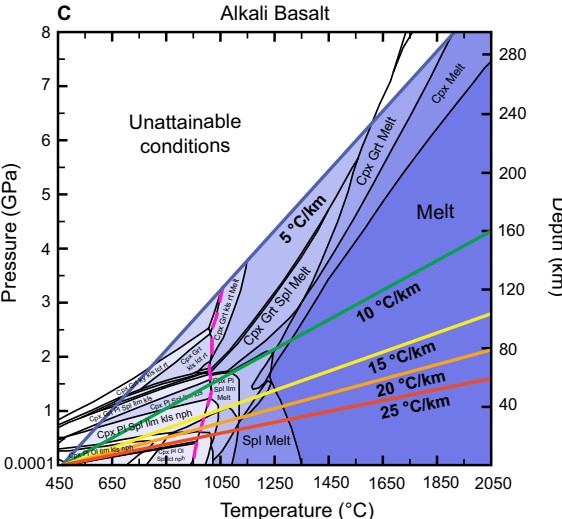

**Fig. 3 | Pseudosections for starting compositions.** Calculated pseudosections for the compositions (Table 2) of peridotite (**A**), basalt (**B**), and alkali basalt (**C**). Small phase fields are not labeled. Variance is shown by shading. Depth scale on the right $y$-axis is estimated using $h = P/\rho g$ (see text for limitations). Solid lines in different colors represent the thermal gradients of 5, 10, 15, 10, and 25 °C/km selected for this study. Conditions to the left of the thermal gradient of 5 °C/km are considered unattainable on Venus although some phase fields were extended into this area. The dashed magenta line represents the solidus. Mineral abbreviations (solid solutions capitalized): coe coesite, Cpx Clinopyroxene, Grt Garnet, Ilm Ilmenite, kls kalsilite, ky kyanite, lct leucite, nph nepheline, Ol Olivine, Opx Orthopyroxene, Pl Plagioclase, qz quartz, rt rutile, Spl Spinel, sti stishovite.

plutonic-squishy lid[67], or may have transitioned between tectonic states over time[68]. A stagnant-lid regime results in a globally thick crust and low surface heat flow, an episodic-lid regime is characterized by variable crustal thickness and heat flow depending on the number and time of overturn events whereas a plutonic-squishy lid regime differs from the other regimes by a thin lithosphere, high conductive heat fluxes, and requires delamination due to the formation of eclogite. Our results show that higher thermal gradients will prevent the formation of eclogite and could therefore limit the mechanism of delamination to colder regions due to a lack of densification and future work will have to address the implications to the plutonic-squishy lid model. Heat flow estimates for Venus show a similar range of variability as Earth[11] reflecting a range of tectonics features, which need to be studied in detail by further models, so we focus on a few examples.

Mead Basin, the largest impact structure on Venus, has relatively low thermal gradients of ≤14 °C/km[33], which would require a locally thick lithosphere analogous to a stagnant lid regime[31,32,69] or episodic stagnant-lid stage[65]. In this thermal state, melt can only be generated in the deep mantle since neither the crust nor the upper mantle reach the solidus within the upper 40–50 km (depending on thermal gradient, Fig. 1). Deep mantle melts or mantle plumes would therefore be the main source for volcanism and new basaltic crust on the surface. Peridotite melt compositions decrease in $SiO_2$ and increase in FeO with increasing pressure[70] and venusian surface rocks with these geochemical signatures could indicate a deep mantle source. The Venera 13 analysis may be an example for melts originating from the deep mantle[15] since it is comparable with silica-undersaturated rocks although there are large uncertainties in the oxide content and a slightly hydrated and substantially carbonated source region may also be required to reproduce its geochemical signatures[71]. However, the identification of deep primary mantle melts by observing their geochemistry from orbit (e.g., emissivity measurements on future Venus missions[72]) will be hampered by ascending magmas being exposed to processes such as magma mixing, crustal assimilation, fractional crystallization, and surface alteration which will alter the geochemistry of primary mantle melts.

For higher thermal gradients of 20–25 °C/km, the predicted maximum crustal thickness will be limited to <30 km (Table 1 and Figs. 1D, E and 2D, E) since the basalt and alkali basalt compositions reach the solidus beyond this depth. Regions with thermal gradients >24 °C/km on Venus are generally associated with coronae, rifts and steep-sided domes[11,12,34], which are represented by thin and weak crust and could therefore be analogous to a mobile-lid regime, or a plutonic-squishy lid regime[67]. High thermal gradients will also result in melting of the peridotite at ~30 km (Figs. 1E and 2E). Low pressure peridotite melts have slightly higher $SiO_2$ and lower FeO than high pressure melts[70]. The Venera 14 and Vega 2 basalts likely derived from an upper mantle origin and could have potentially formed by hot spot or ridge-related volcanism although more precise measurements would be needed to further constrain their formation condition and origin[15].

A transition of global tectonics from a stagnant to a mobile lid regime (globally lower to higher heat flow) would expose large

Venus' global tectonics largely controls global heat flux and magma production; however, its current and past tectonic states are poorly constrained. While observations rule out the presence of a present-day mobile-lid and a classical stagnant-lid state[65], Venus could be in an episodic lid regime with alternating periods of low and high tectonic and magmatic activity[66], in a hybrid regime, such as

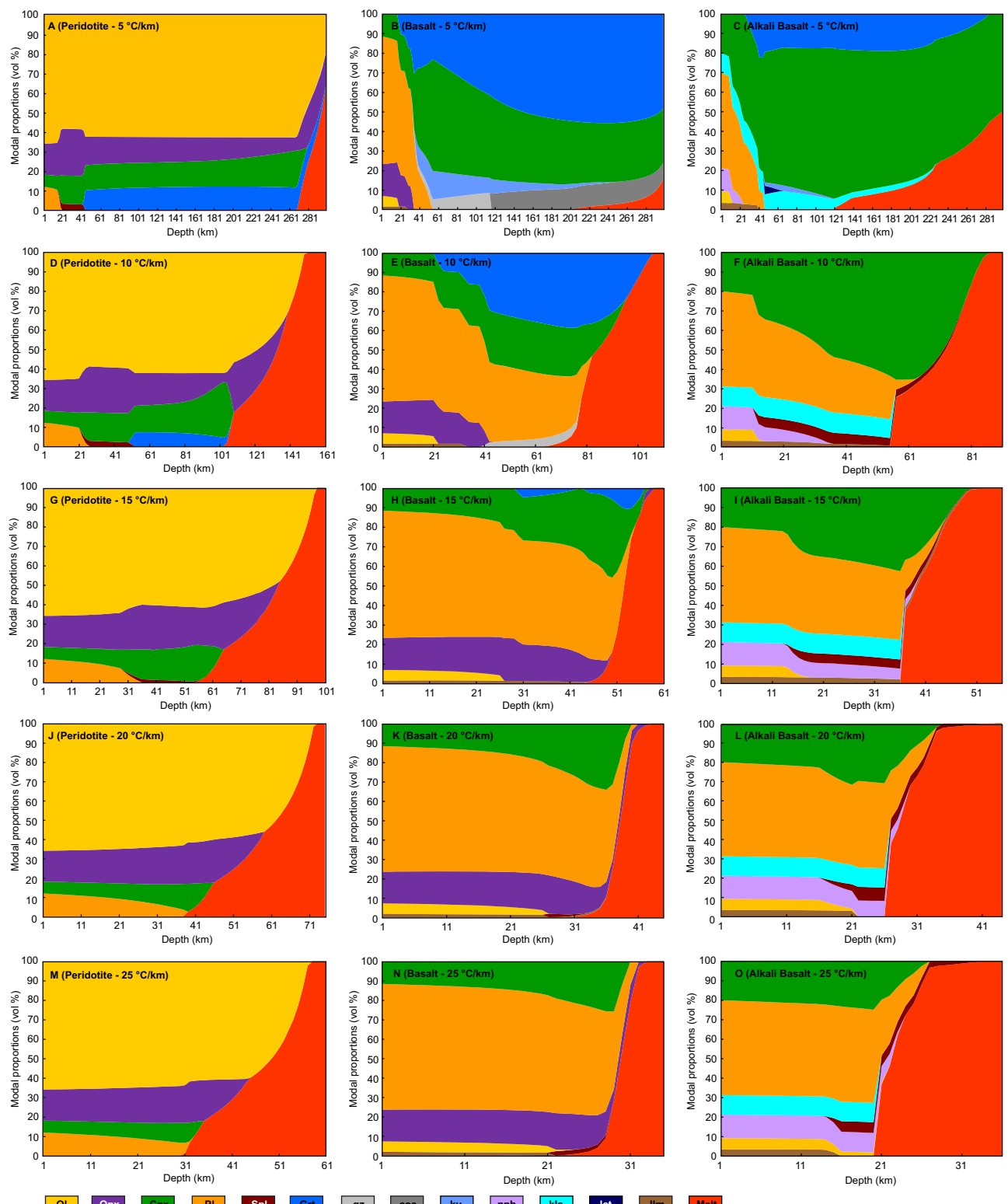

**Fig. 4 | Mineral modes dependent on composition and thermal gradient.** Modal proportions (in vol%) of calculated equilibrium mineral phases for peridotite (**A, D, G, J, M**), basalt (**B, E, H, K, N**), and alkali basalt (**C, F, I, L, O**) and thermal gradients of 5 (**A, B, C**), 10 (**D, E, F**), 15 (**G, H, I**), 20 (**J, K, L**), and 25 (**M, N, O**) °C/km. Phases present as traces of <1 vol% are not shown. See Fig. 3 for mineral abbreviations. The depth (*x*-axis) varies per thermal gradient.

portions of the mafic lower crust (>20 km) to melting conditions and result in the formation of silica-rich melts. A change in global tectonics from mobile lid to stagnant lid (globally higher to lower heat flow) may either preserve a relatively thin crust of ~20 km if there is little formation of new crust by magmatism or result in a thickening of the crust

and subsequent delamination and widespread volcanism consistent with episodic overturns. Convection models show that transitions between tectonic states take more than 1 Ga and are highly energetic with significant amounts of melt generation, which can be spatially discontinuous[68]. As a result, crustal thickness may vary significantly as

**Table 2 | Whole-rock starting compositions given in oxides based on terrestrial peridotite[75], venusian basalt[14,15], and terrestrial alkali basalt[76]**

| Oxide (wt %) | Peridotite | Basalt | Alkali basalt |
|---|---|---|---|
| $SiO_2$ | 44.84 | 48.7 | 47.9 |
| $TiO_2$ | 0.11 | 1.3 | 2.8 |
| $Al_2O_3$ | 3.51 | 17.9 | 18.00 |
| $Fe_2O_3$ | 0.3 | 0.98 | 1.07 |
| FeO | 7.93 | 7.92 | 8.64 |
| MgO | 39.52 | 8.1 | 3.3 |
| CaO | 3.07 | 10.3 | 7.7 |
| $Na_2O$ | 0.3 | 2.4 | 6.0 |
| $K_2O$ | 0.02 | 0.2 | 2.7 |

it could be thickened by increased magmatic activity and underplating in regions of higher heat flow and thinned by delamination in regions of lower heat flow.

## Methods

We used Perple_X[73], a Gibbs free energy minimization software, to calculate equilibrium mineral assemblages for basalt and alkali basalt compositions representative for the venusian crust, and for a peridotite mantle composition. Then we extracted phase proportions and densities for a range of possible lithospheric thermal gradients from 5 °C/km up to 25 °C/km, comparable to estimates for Venus[11,12,32,33] and thermal gradients of Earth's continental crust[74] today. To determine the maximum crustal thickness for each thermal gradient and crustal composition, the following two criteria were used: (1) the depth equivalent to density overturn where the calculated density of the crustal composition becomes denser than the modeled mantle density and subsequently delaminates; and (2) the depth equivalent to the onset of melting of the crust.

### Starting compositions

Three whole-rock compositions were used as starting compositions for our models (Table 2): a peridotite[75], a basalt[14,15], and an alkali basalt[76]. In the absence of a sampled venusian mantle composition, a terrestrial peridotite (a spinel lherzolite from the Kilbourne Hole crater, New Mexico, which is a widely used natural analogue for Earth's upper mantle[75]) was chosen since a peridotitic mantle composition has been inferred for Venus based on the bulk density and moment of inertia of the planet[77]. The basalt composition is derived from analyses of the Venera 14 lander[14,78] and terrestrial analogs for elements not analyzed by the lander[15]. The alkali-rich basalt composition represents a sample from Sverrefjell volcano, Svalbard[76], which is chemically similar to the composition measured by the Venera 13 lander[14] but provides a complete composition without assumptions and uncertainties. Both crustal samples represent the compositional diversity of the venusian crust.

### Phase equilibria modeling

We modeled equilibrium mineral assemblages as a function of pressure, temperature, and composition to infer metamorphism of the venusian crust using the thermodynamic software package Perple_X, which calculates stable phase equilibria by means of Gibbs free energy minimization[73]. An internally consistent thermodynamic data set provided the basis for phase endmember calculations[79,80]. The starting compositions (Table 2) were normalized to 100% by Perple_X at the beginning of each calculation. Models were calculated in the $SiO_2$-$TiO_2$-$Al_2O_3$-$Fe_2O_3$-FeO-MgO-CaO-$Na_2O$-$K_2O$ chemical system. The redox state of the venusian crust and mantle is poorly constrained; however, the FeO/MnO ratio measured at the Venera and Vega landing sites

suggests a rather oxidized surface and interior[81]. $Fe_2O_3$ was defined as a new component in Perple_X by the following relationship: $2FeO + 0.5O_2$ and was then set to 0.3 wt% for the peridotite based on the range of 0.1–0.4 wt% $Fe_2O_3$ measured in terrestrial peridotites[82]. For the basalt and alkali-basalt, $Fe_2O_3$ was calculated as 10% of the total iron content in the bulk rock compositions to reflect an oxidized crust. $Cr_2O_3$, MnO, and $P_2O_5$ were not included in our calculations because of their low abundance and/or limited available solid solution models. All calculations were run without the addition of $H_2O$, $CO_2$, S, and halogens since their abundance in the interior are not known e.g., ref. 15 and our study focuses on constraining the conditions of dry phase transitions and melting before adding additional variables. The following activity solid solution models were used for all starting compositions: olivine, orthopyroxene, clinopyroxene, garnet, spinel, and melt[83]. For the peridotite calculations, we used a plagioclase solid solution from Jennings and Holland[84]. For calculations with the basalt and alkali basalt compositions, we used a ternary plagioclase solid solution[85] and a solid solution model for ilmenite[86]. Rutile, quartz, coesite, stishovite, kyanite, leucite, nepheline, and kalsilite were treated as pure phases. We excluded redundant endmembers of solid solution models that were not used in our calculations and corundum since it is not expected in the compositions considered. Further, feldspathoids were excluded for the basaltic composition since it represents olivine tholeiite.

Phase diagrams for equilibrium conditions were calculated over a pressure and temperature range of 450–2050 °C and 0.0001–8 GPa. We also calculated one-dimensional equilibrium phase diagrams along five geotherms of 5,10,15, 20, and 25 °C/km, based on the range considered in crustal models[33,36,38] and using a surface temperature of 450 °C. Rock densities were extracted using the Perple_X application werami[73]. Where stable, melt was fractionated from the rock before calculating the density. To estimate depth (h) from pressure (P), we used the following equation: h = P/ρg, with a gravity (g) value for Venus of 8.87 m/s² and an average crustal density of 3000 kg/m³. Since density changes as a function of depth, all depth estimates given in this study are approximations e.g., ref. 87.

### Uncertainties and limitations of the model

Uncertainties in our model are mainly derived from uncertainties on physical properties of mineral endmembers in the thermodynamic data set and associated with the formulation of activity-composition relationships of solid solutions[88]. Since these uncertainties affect the phase fields and consequently the density, we assumed an error of 10% on crustal thickness estimates (Table 1). This is reasonable when considering the typically accepted uncertainties reported for pressure (± 2kbar ≈ 5–7 km depth on Earth) and temperature (± 50 °C) estimates determined via phase equilibria modeling due to geologic uncertainty[75]. We justify the exclusion of the oxides $Cr_2O_3$, MnO, and $P_2O_5$ due to their low abundance of <1 wt% in the starting composition and no significant changes in the density profiles are expected. The addition of $Cr_2O_3$ could likely cause a slight increase in the rock density since Cr-bearing phases such as chromite are denser than their Cr-free equivalents. $P_2O_5$ is mainly incorporated in apatite, which could be present as a trace mineral in the basalt and alkali basalt compositions and may therefore also slightly increase the rock density. The effect of MnO, which is usually included in FeO and MgO-bearing minerals (e.g., clinopyroxene, garnet, olivine) should be extremely small. The peridotite melt model is calibrated up to ~5 GPa and that for basaltic melts up to ~3 GPa[83] and the melt curve shown above these pressures is an extrapolation. There is currently no calibration for the alkali basalt although our melt model includes K end members. The results shown for the alkali basalt may therefore also show larger uncertainties. We also made the simplification of not including any volatiles since their abundance in the subsurface is currently poorly constrained and the relatively young venusian crust, which is believed to have formed by

significant volcanic activity[89–92] is likely depleted in volatiles ($H_2O$, $CO_2$, S) due to degassing. However, partial melting of the crust and mantle is not expected to result in the interior to be entirely volatile-free and a fraction of the initial volatile content will be likely retained in the interior[10]. Therefore, the influence of volatiles on phase transitions and melting should be considered in future studies, particularly since even low abundances of $H_2O$ and $CO_2$ can significantly influence melting temperatures of peridotite and basaltic compositions e.g., refs. [93,94].

## Data availability

Model input data including starting compositions and solid solution models are provided in this paper. The model data generated in this study are provided in Supplementary Data 1.

## Code availability

All models have been calculated with the freely available code Perple_X, available for download here: https://www.perplex.ethz.ch/

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

## Acknowledgements

This work was partially funded by the Lunar and Planetary Institute's Summer Internship Program. J.S. acknowledges the Research England Expanding Excellence in England (E3) fund (grant code 124.18). J.F. and J.G. thank partial support from NASA's Planetary Science Division Research Program, through ISFM. J.F. acknowledges partial support from DAVINCI. M.W. acknowledges partial support through NASA's Solar System Workings program grant 80NSSC23K0167, and additional support through NASA Cooperative Agreement 80NSSC24M008 to the Lunar and Planetary Institute.

## Author contributions

Conceptualization: J.F., J.S., J.G. Methodology: J.S., J.G., N.C. Investigation: J.S., J.F., N.C., J.G., M.W. Visualization: J.S., N.C. Supervision: J.S., J.F., J.G. Writing—original draft: J.S., J.F., M.W., J.G., N.C. Writing—review & editing: J.S., J.F., M.W., J.G.

## Competing interests

The authors declare no competing interests.
