## [Transparent Peer Review file · Nature Communications]

Metamorphism of Venus as driver of crustal thickness and recycling

Corresponding Author: Dr Julia Semprich

Version 0:

Reviewer comments:

Reviewer #1

(Remarks to the Author)

I enjoyed reading this manuscript, which uses phase equilibrium modelling of primitive crust and lithospheric mantle compositions to calculate the density structure and thereby gravitational stability and thickness of the crust on Venus. It is a very powerful approach that I think we can have some trust in. However, while this is very interesting study that I would like ultimately to see published, there are a number of things that concern me. Most importantly:

Phase equilibria:

With reference to the phase equilibrium modelling and Fig. 3, the phase diagram for the peridotite is fine. However, the melt model for the basalts has NOT been calibrated about ~2 GPa. This is clearly indicated by the back bending of the high pressure solidus (dotted pink line) to ~4.8 GPa at 450 C. Does that seem reasonable? You can ONLY show and interpret the phase equilibria for the basalts (Fig. 3b,c) to 2 GPa max. It doesn't affect much of what you later write, but it is wrong to show it. I also worry that the Green et al. melt model used is not calibrated for alkali basalts, a project I believe Owen Weller and others at Cambridge (and elsewhere) have been working on. Personally, I would model only the venusian basalt and it's (assumed) lithospheric mantle residue.

With respect to the bulk compositions modelled, the sentence "For the basalt and alkali-basalt, Fe₂O₃ was calculated as 10% of the total iron content in the bulk rock compositions." needs support. What is based on? If on Earth's mantle, then is that a reasonable comparison? The statement (l.787) "feldspathoids were excluded for the basaltic composition" also needs support, and the same elsewhere.

References:

The theme of the omission of key references gets personal in the introduction, where the authors lay out the utility of phase equilibria in constraining models of lithospheric evolution. It is one I used myself in a 2014 paper...

Johnson, T.E., Brown, M., Kaus, B.J. and VanTongeren, J.A., 2014. Delamination and recycling of Archaean crust caused by gravitational instabilities. *Nature Geoscience*, 7(1), pp.47-52.
...which modelled the gravitational stability of Archean crust on Earth.

We did similar for the Moon in 2021.

Johnson, T.E., Morrissey, L.J., Nemchin, A.A., Gardiner, N.J. and Snape, J.F., 2021. The phases of the Moon: Modelling crystallisation of the lunar magma ocean through equilibrium thermodynamics. *Earth and Planetary Science Letters*, 556, p.116721.

The manuscript reads like this is a new approach, but it isn't.

Metamorphic terminology:

Throughout there is a fundamental misunderstanding of the meaning of a metamorphic facies. As the authors state, an

anhydrous rock cannot grow amphibole or any other hydrous mineral. That does not mean it cannot be metamorphosed to amphibolite (or greenschist) facies conditions. Similarly, the use of hornfels and hornfels facies is NOT helpful. These terms have a very specific (contact metamorphic) inference that is at best misleading. Here, you are simply talking about shallow (low P) and deep (higher P) crust, one which contains garnet and another that doesn't. While Figure 1 is pretty, it would be MUCH more useful to see some of the important phase transitions on it. With these and other aspects, including the discussion of what to call rocks with lots of garnet and clinopyroxene in, I would urge the authors to read Johnson et al. (2014).

Geodynamics:

The authors talk a lot about delamination of crust as a limiting factor, but there is a difference between crust becoming negatively buoyant and crust overturning/delaminating/subducting/whatever. An apple placed on a table is negatively buoyant and wants to sink, but it doesn't because the table is too strong. Whether or not negatively buoyant crust sinks relates mostly to the viscosity of the underlying mantle, itself a function of T, H₂O content, melt fraction, blah blah. Numerical modelling is a good way to look at this problem, as done by Johnson et al. (2014) and others. These 2d models, if sufficiently simple, can run on decent home computers. Persuading somebody to help you with this, in addition to paying attention to the above, would take this to the next level, and permit a few firmer constraints on the possible implications of the phase equilibria to understanding the tectonics of Venus.

In case it's helpful I will try and remember to attach an edited Word doc, where I have edited things (using Track Changes) for clarify and on which I have provided a number of additional comments and suggestions. Sorry, I did not finish the discussion.

Good luck with it,

Tim Johnson

Reviewer #2

(Remarks to the Author)

Review by Ernst of "Metamorphism of Venus: Driver of Crustal Thickness, Recycling, and Volcanism" by Semprich et al.

GENERAL COMMENTS: I am really pleased to see this manuscript which is very solid piece of work for which I have only minor suggestions.

There has been a need for such a study to provide constraints on Venusian crustal thickness and composition, incorporating the factor that is so dramatically different than on Earth—surface temperatures of 450 degree C per km. The background information sections are robust and the manuscript is well written.

The modelling seems very thorough and logical (although the details of such modelling are outside my main expertise). Their approach incorporates metamorphic thermodynamic approach (Gibbs free energy minimization) – using three plausible geothermal gradients (5, 10 and 25 degrees C per km, which are arguably present in different parts of Venus). The paper provides basic constraints on the thickness of crust. Notably, under a high geothermal gradient the thickness of the crust is limited by melting at the base. For the lowest geothermal gradient, there is a potential for delamination related to formation of eclogite. Based on their modelling they infer a range of possible crustal thicknesses between 20 and 65 km. they note that for thicknesses observed greater than 65 km, this would indicate some silicic unit component to the crust.

This mention of >65 km requiring silicic units does raise an interesting question which is really outside the scope of their research. The stratigraphically oldest units, tesserae (which also tend to be associated with thick crust) are exposed over about 8% of Venus, but tesserae may have much wider extent under basaltic plains units (e.g. Ivanov and Head, 2007? Or 1997?). Some studies are suggesting that tesserae may have a silicic component (e.g. Gilmore, and I think there is a new paper by Paul Byrne that mentions this)—so perhaps there are regions of Venus with crustal thicknesses less than 65 km for which silicic crust may be present. Adding a modelling component that incorporates some silicic crust is really outside the scope of this paper, but it would be useful for the authors to say a few things about this possibility of silicic units in tessera being more widespread.

DETAILED COMMENTS;

Line 28: missing reference on evidence for young volcanism--- Shalygin et al. 2015

Lines 44: could add Way et al. 2022 (which indicates the potential of volcanism in causing a "Great Climate Transition" on Venus

Also could add Khawja et al. 2020 (which proposed that the oldest terrains tesserae may preserve evidence of fluvial erosion- consistent with tesserae preserving a protracted geological history during the earlier more habitable climate proposed by Way et al. 2016 for Venus

Line 59: perhaps add Elkin Tanton (e.g. Elkins-Tanton et al. 2007) on delamination

Lines 190-192: you mention the possibility of more silicic crust involved where the crustal thickness is greater than 65 km. However, there are papers by Ivanov and Head and others suggesting that tesserae terrain (which covers about 8% of Venus could be widespread underlying underlie basaltic units—and there has been a suggest for tesserae having a significant silicic component.

Could you add some text somewhere in the manuscript acknowledging this possibility

Lines 327-331: you explain your procedure for determining the Fe₂O₃ amount---- Can you provide a bit more of rationale for this selection

REFERENCES TO ADD:

Shalygin, E. V., et al. (2015), Active volcanism on Venus in the Ganiki Chasma rift zone, *Geophys. Res. Lett.*, 42, 4762–4769, doi:10.1002/2015GL064088.

Khawja, et al. (2020). Tesserae on Venus may preserve evidence of fluvial erosion. *Nature Communications*, 11, article no 5789. <https://doi.org/10.1038/s41467-020-19336-1>

Way, M.J., et al. (2022). Large scale volcanism and the heat-death of terrestrial worlds. *The Planetary Science Journal*, 3:92 (11pp), <https://doi.org/10.3847/PSJ/ac6033>

Elkins-Tanton, et al. (2007). Continental magmatism, volatile recycling, and a heterogeneous mantle caused by lithospheric gravitational instabilities. *Journal of Geophysical Research*, 112: B03405

Reviewer #3

(Remarks to the Author)

The authors investigate the metamorphism of Venus' crust, with a particular focus on phase transitions and the onset of melting under different lithospheric thermal gradients. They employ a phase equilibria modeling approach to assess the effects of pressure and temperature on the mineralogy of dry basalt, alkali basalt, and peridotite mantle compositions. By calculating phase equilibria as a function of pressure, temperature, and composition, the authors demonstrate how stable mineral assemblages and corresponding rock densities vary gradually with depth, rather than as an average. The study also compares the calculated crust and mantle densities to identify where the crust may become denser than the underlying mantle. Additionally, the authors correlate their findings with estimates of crustal thickness to predict regions on Venus where crustal melting or recycling through delamination is likely to occur. Given the growing scientific interest in Venus and the need for a deeper understanding of its crustal compositions and geodynamics, this paper is particularly timely and relevant to the readership of a high-impact journal such as *Nature Communications*.

In my opinion, this manuscript is interesting, well-written, and structured, and the science is sound. I have only a few minor concerns that I do believe the authors should address before this paper can be published. These minor concerns converge towards the better acknowledgment of other relevant interpretations of geodynamics for Venus and some other needed discussions of some other relevant literature. I am confident the authors can address these comments (or revoke them if they disagree with reasoning) relatively easily.

- Crustal thickness linked to thermal gradient and global tectonics. The connection between thermal gradients and tectonic settings lacks sufficient elaboration. The authors predominantly consider a binary global tectonic framework of “mobile lid” versus “stagnant lid” regimes, citing mostly their own work on geodynamic regimes they consider appropriate to Venus. This approach overlooks the existence of alternative (or, adapted) global tectonic models for present-day Venus, such as the “sluggish” or “squishy” lid regimes, which suggest partial mobility and laterally variable deformation due to magmatic-induced weakening. These alternative models are particularly relevant to the brief discussion in lines 266-273 and should be acknowledged. In particular, the statement in lines 270-273 (“as a result, crustal thickness may vary significantly as it could be thickened by increased magmatic activity and underplating in regions of higher heat flow and thinned by delamination in regions of lower heat flow”) appears to describe melt generation primarily as a consequence of long-term tectonic regime transitions, with a self-citation. However, there is substantial existing research on these topics that should be considered, such as the work by Byrne et al. (2021 <https://www.pnas.org/doi/10.1073/pnas.2025919118>), Lourenco et al. (2020 <https://agupubs.onlinelibrary.wiley.com/doi/10.1029/2019GC008756>) (as described in the review by Rolf et al. (2022, <https://asu.elsevierpure.com/en/publications/dynamics-and-evolution-of-venus-mantle-through-time>), which the authors are familiar with).

- Moreover, surface heat flow estimates for Venus, despite their inherent limitations, are available and show variability across the planet, as noted by Smrekar et al. (2023) (cited as #10 in the manuscript). It would be beneficial for the authors to briefly contextualize their discussion on thermal gradients within the range of these varying heat flow values across different tectonic structures on the planet, not just stagnant/mobile lid tectonic regimes (the Mead Basin is discussed, but it is not so clearly acknowledged that the wealth of different tectonic structures on Venus, e.g. rifts, coronae, hotspot swells, tessera highlands, may all have different thermal gradient within one geodynamic regime)

- Phase equilibria have been made for Vega 2, Venera 13, and Venera 14 Venus rock measurements in Chen et al. (2022: <https://www.nature.com/articles/s41467-022-35304-3>) (see their main manuscript but also supplementary material). This is highly relevant past work that to which results should be (briefly) compared in this manuscript, as well as the conclusions made by those authors (supposedly “light” Venus lithosphere).

• Average densities of the basaltic crust under different temperature gradients: The high crustal density values are interesting as they indeed contrast to values commonly used in geophysical/geodynamic models. I have a few minor concerns related to this paragraph:

- Line 146 should be revised, as it references "geophysical and geodynamic models," yet only cites geophysical studies for crustal thickness values (#34-36) without including any geodynamic modeling studies that account for crustal compositions.
- Indeed, some models like Adams et al. (2022, https://agupubs.onlinelibrary.wiley.com/doi/epdf/10.1029/2022JE007460?saml_referrer) use 2900 kg/m³ as Venus crustal density; although a set of other tectonic models (Gülcher et al. (2020; 2023) and Schools and Smrekar (2024 <https://www.sciencedirect.com/science/article/pii/S0012821X24000761>) use equations-of-state that lead to crustal density values around 2900-2970 kg/m³, using the following equation of state: $\rho = \rho_0 [1 - \alpha(T - 298)] [1 + \beta(P - 0.1)]$, (T in Kelvin; P in MPa) with thermal expansion $\alpha = 3 \times 10^{-5} \text{ K}^{-1}$, compressibility $\beta = 10^{-5} \text{ MPa}^{-1}$ and $\rho_0 = 3000 \text{ kg/m}^3$ as the rock density at standard temperature and pressure. Another recent tectonic modeling work by Regorda et al. (2023 <https://agupubs.onlinelibrary.wiley.com/doi/abs/10.1029/2022JE007588>) Therefore, the values mentioned in the text (2800-2900 kg/m³) are not exactly covering the range of geodynamic models.

• Methods:

o this seems very obvious, but I see nowhere stated what the surface temperature used is for the thermal profiles with the gradients? I assume 460°C based on line 26? It would be good to state in Methods regardless.

o Lines 350: I am under the impression the gravity value for Venus is 8.87 m/s², and not 8.77 (<https://nssdc.gsfc.nasa.gov/planetary/factsheet/venusfact.html>). Did you really use 8.77 and if so, how is this motivated? I am of the opinion that the calculations should be done with 8.87 m/s² for accuracy.

Version 1:

Reviewer comments:

Reviewer #1

(Remarks to the Author)

It's always nice to devote ones time to doing a review in good faith, signing your name to that review, then receive snippy and and time rude responses from the authors. This is hardly uncommon from a planetary community who seem to begrudge any comments from geologists. Hey ho.

The point regarding references was meant to credit those very few studies that have used a phase equilibrium modelling approach used by the authors on other differentiated silicate bodies, even if they were done by geologists. If the authors don't see the relevance and/or that these papers should be referenced, then fine. I thank them for reluctantly adding a reference to our 2014 paper on delamination, but they should feel free to remove it if they need to trim the bibliography.

I am not an expert on Venus, capitalised or not, although I do have plenty of idea to explain some of the features it preserves. However, I do have some expertise in metamorphism and some documented experience of how to express ideas to a broad readership. Regarding metamorphic terminology, if they think a broad readership will understand phrases like 'hornfels and hornfels facies' better than my suggested modifications, then they should revert back to the original. It is their paper upon which I am only providing advice.

Regarding the geodynamics section, rather than not having read the discussion, perhaps I put together my comments piecemeal and, what with other time pressures, did not find time to cross check everything before submitting my review as close to the deadline (albeit over it) as I could manage. I can only offer my apologies.

That the authors think they have "we have made it clear in the manuscript that our conclusions are based on petrological constraints and not on additional parameters" does not prevent them from discussing some of those parameters in a section that is specifically designed for such things. If they don't want to then they don't have to.

As before, this is a nice study that should be published.

Tim Johnson, Perth, January 2025

Reviewer #4

(Remarks to the Author)

Paper summary and key points of interest: The authors investigate the metamorphism of Venus' crust, with a particular focus on phase transitions and the onset of melting of mafic rocks on Venus' volcanic plains (which cover the majority of the planet's surface) under different lithospheric thermal gradients that could represent different geodynamic settings. They employ a phase equilibria modeling approach to assess the effects of pressure and temperature on the mineralogy of dry basalt, alkali basalt, and peridotite mantle compositions. Ultimately, they present their calculations of the density structures under various (thermal, compositional) conditions, and assess the possibility of gravitational instabilities, onsets of melting, and ultimately, the maximum possible thickness of the mafic crust. Several findings, such as that delamination may not always occur at the phase boundary, the commonly made assumption of a linear increase in density from gabbro to eclogite is inadequate, and that crustal thickness may be limited by melting rather than delamination under 'thin' lithosphere

conditions, have significant implications on various Venus science studies.

Assessment: I enjoyed reading this revised manuscript. I think the authors have addressed the wide range of points made by all three reviewers, pointing out very different aspects, adequately.

I am happy that the authors have decided to specifically clarify the applicability of their results (i.e., to stable mafic crust and not to other tectonic/dynamic areas), to include a wider discussion of the available concepts of Venus global tectonics, and clarifies some key future work aspects out-of-scope of this paper (e.g. relevance to perhaps silica-rich compositions in tesserae). Overall, I think the paper will be a good contribution to Nature Communication and of high interest to the journals readership and the (Venus) science community.

REVIEWER COMMENTS

We thank the reviewers for their comments and have revised our manuscript as described in our responses below. Line references should match those in the revised version.

Reviewer #1:

I enjoyed reading this manuscript, which uses phase equilibrium modelling of primitive crust and lithospheric mantle compositions to calculate the density structure and thereby gravitational stability and thickness of the crust on Venus. It is a very powerful approach that I think we can have some trust in. However, while this is very interesting study that I would like ultimately to see published, there are a number of things that concern me. Most importantly:

Phase equilibria:

With reference to the phase equilibrium modelling and Fig. 3, the phase diagram for the peridotite is fine. However, the melt model for the basalts has NOT been calibrated about ~2 GPa. This is clearly indicated by the back bending of the high pressure solidus (dotted pink line) to ~4.8 GPa at 450 C. Does that seem reasonable? You can ONLY show and interpret the phase equilibria for the basalts (Fig. 3b,c) to 2 GPa max. It doesn't affect much of what you later write, but it is wrong to show it. I also worry that the Green et al. melt model used is not calibrated for alkali basalts, a project I believe Owen Weller and others at Cambridge (and elsewhere) have been working on. Personally, I would model only the venusian basalt and it's (assumed) lithospheric mantle residue.

Response: We would like to point out that we used the melt model from Holland et al., 2018 and not the one for metabasic rocks described in Green et al., 2016. This is stated in the phase equilibria section of the Methods. The newer model is calibrated up to ~5 GPa for peridotite (although extrapolation to higher pressure for peridotite seems to be reasonable as R#1 suggests) and up to 3 GPa for eclogite (see Fig. 6 in Holland et al. 2018). It is therefore not clear why we can only show and interpret phase equilibria up to a maximum of 2 GPa. We do acknowledge that the phase diagrams in Fig. 3 showed phase fields in the region to the left of the thermal gradient of 5 °C/km, which we assumed were obviously neither attainable for Venus, nor relevant for the manuscript. We have provided a new Fig 3, where this area is now clearly labelled as representing 'unattainable conditions' and have changed the figure caption accordingly (lines 629-631). We have also added the calibration limits in the 'Uncertainties and limitations of the model' subsection of the Methods (lines 346-348). The melt model from Holland et al., 2018 also includes a K end member, which is why we have kept the results for alkali basalt in the manuscript and added additional caveats to the same section (lines 348-350).

With respect to the bulk compositions modelled, the sentence "For the basalt and alkali-basalt, Fe₂O₃ was calculated as 10% of the total iron content in the bulk rock compositions." needs support. What is based on? If on Earth's mantle, then is that a reasonable comparison? The statement (l.787) "feldspathoids were excluded for the basaltic composition" also needs support, and the same elsewhere.

Response: We have added a sentence to clarify the issue with poorly constrained redox state including a reference that summarizes the limited data with the prospect of a relatively oxidized surface and interior (lines 302-304 and 308). Based on these inferred assumptions, we can assume that Fe_2O_3 is not negligible, but we would also like to point out that variations in a few percent of Fe_2O_3 will overall not add significant uncertainties to the phase diagrams or extracted densities. We also added more information to support the exclusion of feldspathoids (line 321).

References:

The theme of the omission of key references gets personal in the introduction, where the authors lay out the utility of phase equilibria in constraining models of lithospheric evolution. It is one I used myself in a 2014 paper...

Johnson, T.E., Brown, M., Kaus, B.J. and VanTongeren, J.A., 2014. Delamination and recycling of Archaean crust caused by gravitational instabilities. *Nature Geoscience*, 7(1), pp.47-52.

...which modelled the gravitational stability of Archean crust on Earth.

We did similar for the Moon in 2021.

Johnson, T.E., Morrissey, L.J., Nemchin, A.A., Gardiner, N.J. and Snape, J.F., 2021. The phases of the Moon: Modelling crystallisation of the lunar magma ocean through equilibrium thermodynamics. *Earth and Planetary Science Letters*, 556, p.116721.

The manuscript reads like this is a new approach, but it isn't.

Response: According to the author guidelines for Nature Communications, references should not exceed 70 (<https://www.nature.com/ncomms/submit/article>). The manuscript is already slightly over this limit due to the large range of topics covered. It seems that opinions differ significantly on what 'key' references need to be included, especially when looking at the suggestions from R#2 and R#3. We have added the first publication by R#1 (line 59), but we have some difficulties seeing the relevance of the crystallization of the lunar magma ocean to a planet the size of Venus discussing phase equilibria of the solid crust and mantle.

It is not clear where the manuscript seems to read like this is a new approach since there are clearly references to delamination and the application of phase equilibria in the introduction. The new aspect of this manuscript is the application of phase transitions to **Venus** which R#1 seems to acknowledge in their introductory remarks.

Metamorphic terminology:

Throughout there is a fundamental misunderstanding of the meaning of a metamorphic facies. As the authors state, an anhydrous rock cannot grow amphibole or any other hydrous mineral. That does not mean it cannot be metamorphosed to amphibolite (or greenschist) facies conditions. Similarly, the use of hornfels and hornfels facies is NOT helpful. These terms have a very specific (contact metamorphic) inference that is at best misleading. Here, you are simply talking about shallow (low P) and deep (higher P) crust, one which contains garnet and another that doesn't. While Figure 1 is pretty, it would be MUCH more useful to see some of the important phase transitions on it. With these and other aspects, including

the discussion of what to call rocks with lots of garnet and clinopyroxene in, I would urge the authors to read Johnson et al. (2014).

Response: What R#1 calls a fundamental misunderstanding was, in fact, an intentional decision on our part. We have taken the general guidelines for Nature Communications seriously in writing for a 'specific research community'. In our case, this is the Venus community, which includes researchers from diverse fields, many of whom are not familiar with the concept of metamorphic facies. Years of experience presenting phase diagrams within this community has shown that a greenschist-facies rock is mostly associated with the presence of water, and we intentionally wanted to prevent these misunderstandings. It seems that our solution of trying to be more inclusive has caused some other issues, so we have decided to remove all references to rock types and metamorphic facies in Fig. 1 (+ caption) and the manuscript (lines 30, 100, 101, 108-109, 110-111). We replaced hornfels with metagabbro and granulite with garnet metagabbro in Fig. 1 and changed the figure caption (line 610) accordingly. Due to the above-mentioned diversity in the Venus research community, Fig. 1 is intentionally a summary and overview, and we believe that phase transitions will add unnecessary complexity, particularly because these details are shown in Fig. 3 and Fig. 4.

Additional note: Venus is very likely still volcanically active and most of the surface is less than 1 Ga old due to resurfacing (although there is currently no consensus if this was catastrophic or gradual). This will expose a significant part of the surface and the upper crust to contact metamorphism and therefore hornfels-facies. Without the presence of plate tectonics or water these mineral assemblages would likely be preserved until reaching the garnet stability field.

Geodynamics:

The authors talk a lot about delamination of crust as a limiting factor, but there is a difference between crust becoming negatively buoyant and crust overturning/delaminating/subducting/whatever. An apple placed on a table is negatively buoyant and wants to sink, but it doesn't because the table is too strong. Whether or not negatively buoyant crust sinks relates mostly to the viscosity of the underlying mantle, itself a function of T, H₂O content, melt fraction, blah blah. Numerical modelling is a good way to look at this problem, as done by Johnson et al. (2014) and others. These 2d models, if sufficiently simple, can run on decent home computers. Persuading somebody to help you with this, in addition to paying attention to the above, would take this to the next level, and permit a few firmer constraints on the possible implications of the phase equilibria to understanding the tectonics of Venus.

Response: This may be because R#1 did not finish reading the discussion, but the crustal thickness is limited by melting for most thermal gradients and delamination is only likely in cases of extremely low thermal gradients, which may not be very common on Venus. We think that we have made it clear in the manuscript that our conclusions are based on petrological constraints and not on additional parameters. We do agree that numerical modeling is the next step, but we think that the integration of the petrological data into 3D spherical models would be a much better approach, particularly since MBW has done a considerable amount of work in this area.

In case it's helpful I will try and remember to attach an edited Word doc, where I have edited things (using Track Changes) for clarify and on which I have provided a number of additional comments and suggestions. Sorry, I did not finish the discussion.

Good luck with it,

Tim Johnson

Response: Thanks. Unfortunately, we could not locate an additional file.

Reviewer #2 (Remarks to the Author):

Review by Ernst of “Metamorphism of Venus: Driver of Crustal Thickness, Recycling, and Volcanism” by Semprich et al.

GENERAL COMMENTS: I am really pleased to see this manuscript which is very solid piece of work for which I have only minor suggestions.

There has been a need for such a study to provide constraints on Venusian crustal thickness and composition, incorporating the factor that is so dramatically different than on Earth—surface temperatures of 450 degree C per km. The background information sections are robust and the manuscript is well written.

The modelling seems very through and logical (although the details of such modelling are outside my main expertise). Their approach incorporates metamorphic thermodynamic approach (Gibbs free energy minimization) – using three plausible geothermal gradients (5, 10 and 25 degrees C per km, which are arguably present in different parts of Venus). The paper provides basic constraints on the thickness of crust. Notably, under a high geothermal gradient the thickness of the crust is limited by melting at the base. For the lowest geothermal gradient, there is a potential for delamination related to formation of eclogite. Based on their modelling they infer a range of possible crustal thicknesses between 20 and 65 km. they note that for thicknesses observed greater than 65 km, this would indicate some silicic unit component to the crust.

This mention of >65 km requiring silicic units does raise an interesting question which is really outside the scope of their research. The stratigraphically oldest units, tesserae (which also tend to be associated with thick crust) are exposed over about 8% of Venus, but tesserae may have much wider extent under basaltic plains units (e.g. Ivanov and Head, 2007? Or 1997?). Some studies are suggesting that tesserae may have a silicic component (e.g. Gilmore, and I think there is a new paper by Paul Byrne that mentions this)—so perhaps there are regions of Venus with crustal thicknesses less than 65 km for which silicic crust may be present. Adding a modelling component that incorporates some silicic crust is really outside the scope of this paper, but it would be useful for the authors to say a few things about this possibility of silicic units in tessera being more widespread.

Response: Thank you for the comments. We are very aware of the research related to tesserae since we currently have a manuscript ready for submission, which is focused specifically on the composition to tesserae. We think that this topic is much better covered by itself than adding additional compositions to this manuscript.

DETAILED COMMENTS;

Line 28: missing reference on evidence for young volcanism--- Shalygin et al. 2015

Response: Thanks for pointing this out. Reference added and text slightly modified (line 28).

Lines 44: could add Way et al. 2022 (which indicates the potential of volcanism in causing a “Great Climate Transition” on Venus

Also could add Khawja et al. 2020 (which proposed that the oldest terrains tesserae may preserve evidence of fluvial erosion- consistent with tesserae preserving a protracted geological history during the earlier more habitable climate proposed by Way et al. 2016 for Venus

Response: We agree that these suggestions support the reference list from various angles and have added them (line 43).

Line 59: perhaps add Elkin Tanton (e.g. Elkins-Tanton et al. 2007) on delamination

Response: Added (line 59).

Lines 190-192: you mention the possibility of more silicic crust involved where the crustal thickness is greater than 65 km. However, there are papers by Ivanov and Head and others suggesting that tesserae terrain (which covers about 8% of Venus could be widespread underlying underlie basaltic units—and there has been a suggest for tesserae having a significant silicic component.

Could you add some text somewhere in the manuscript acknowledging this possibility

Response: We agree that this is an important topic that needs to be addressed but as mentioned above, we already have a manuscript ready to focus only on the composition of tesserae and have therefore intentionally not included a lot of discussion on this topic. Looking at the comment from R#3 about heat flow and tectonic features, we acknowledge that we should have made it clear that our current approach is relevant for stable mafic crust and not the range of tectonic features for Venus (which is beyond the scope of this manuscript). We have added a sentence at the beginning of the results section to determine the applicability of our results (lines 91-93) and another sentence in the discussion to add the relevance of silica-rich compositions to tesserae (lines 174-175).

Lines 327-331: you explain your procedure for determining the Fe₂O₃ amount---- Can you provide a bit more of rationale for this selection

Response: We have amended the text to reflect the lack of data on redox state and the assumptions behind using this value (see also response to comments from R#1; lines 302-304).

REFERENCES TO ADD:

Shalygin, E. V., et al. (2015), Active volcanism on Venus in the Ganiki Chasma rift zone, *Geophys. Res. Lett.*, 42, 4762–4769, doi:10.1002/2015GL064088.

Khawja, et al. (2020). Tesserae on Venus may preserve evidence of fluvial erosion. *Nature*

Communications, 11, article no 5789. <https://doi.org/10.1038/s41467-020-19336-1>

Way, M.J., et al. (2022). Large scale volcanism and the heat-death of terrestrial worlds. *The Planetary Science Journal*, 3:92 (11pp), <https://doi.org/10.3847/PSJ/ac6033>

Elkins-Tanton, et al. (2007). Continental magmatism, volatile recycling, and a heterogeneous mantle caused by lithospheric gravitational instabilities. *Journal of Geophysical Research*, 112: B03405

Reviewer #3 (Remarks to the Author):

The authors investigate the metamorphism of Venus' crust, with a particular focus on phase transitions and the onset of melting under different lithospheric thermal gradients. They employ a phase equilibria modeling approach to assess the effects of pressure and temperature on the mineralogy of dry basalt, alkali basalt, and peridotite mantle compositions. By calculating phase equilibria as a function of pressure, temperature, and composition, the authors demonstrate how stable mineral assemblages and corresponding rock densities vary gradually with depth, rather than as an average. The study also compares the calculated crust and mantle densities to identify where the crust may become denser than the underlying mantle. Additionally, the authors correlate their findings with estimates of crustal thickness to predict regions on Venus where crustal melting or recycling through delamination is likely to occur. Given the growing scientific interest in Venus and the need for a deeper understanding of its crustal compositions and geodynamics, this paper is particularly timely and relevant to the readership of a high-impact journal such as *Nature Communications*.

In my opinion, this manuscript is interesting, well-written, and structured, and the science is sound. I have only a few minor concerns that I do believe the authors should address before this paper can be published. These minor concerns converge towards the better acknowledgment of other relevant interpretations of geodynamics for Venus and some other needed discussions of some other relevant literature. I am confident the authors can address these comments (or revoke them if they disagree with reasoning) relatively easily.

- Crustal thickness linked to thermal gradient and global tectonics. The connection between thermal gradients and tectonic settings lacks sufficient elaboration. The authors predominantly consider a binary global tectonic framework of “mobile lid” versus “stagnant lid” regimes, citing mostly their own work on geodynamic regimes they consider appropriate to Venus. This approach overlooks the existence of alternative (or, adapted) global tectonic models for present-day Venus, such as the “sluggish” or “squishy” lid regimes, which suggest partial mobility and laterally variable deformation due to magmatic-induced weakening. These alternative models are particularly relevant to the brief discussion in lines 266-273 and should be acknowledged. In particular, the statement in lines 270-273 (“as a result, crustal thickness may vary significantly as it could be thickened by increased magmatic activity and underplating in regions of higher heat flow and thinned by delamination in regions of lower heat flow”) appears to describe melt generation primarily as a consequence of long-term tectonic regime transitions, with a self-citation. However, there is substantial existing research on these topics that should be considered, such as the work by Byrne et al. (2021 <https://www.pnas.org/doi/10.1073/pnas.2025919118>), Lourenco et al. (2020 <https://agupubs.onlinelibrary.wiley.com/doi/10.1029/2019GC008756>) (as described in

the review by Rolf et al. (2022, <https://asu.elsevierpure.com/en/publications/dynamics-and-evolution-of-venus-mantle-through-time>), which the authors are familiar with).

Response: This is a good point and we actually had a more detailed discussion in an earlier version of the manuscript, which we have taken back in together with a discussion of possible tectonic regimes for Venus citing most of the suggested work (paragraph in the discussion, lines: 216-230, and additions in lines 233, 252).

- Moreover, surface heat flow estimates for Venus, despite their inherent limitations, are available and show variability across the planet, as noted by Smrekar et al. (2023) (cited as #10 in the manuscript). It would be beneficial for the authors to briefly contextualize their discussion on thermal gradients within the range of these varying heat flow values across different tectonic structures on the planet, not just stagnant/mobile lid tectonic regimes (the Mead Basin is discussed, but it is not so clearly acknowledged that the wealth of different tectonic structures on Venus, e.g. rifts, coronae, hotspot swells, tessera highlands, may all have different thermal gradient within one geodynamic regime)

Response: We agree that this is an important issue that should be mentioned in the manuscript, and we have added a sentence about the variability of surface heat flow and tectonic settings in the above-mentioned paragraph (lines 216-230). However, our results will not be applicable to all observed features on Venus (particularly not tesserae – see comment to R#2 above) and this will have to be addressed in future work.

- Phase equilibria have been made for Vega 2, Venera 13, and Venera 14 Venus rock measurements in Chen et al. (2022: <https://www.nature.com/articles/s41467-022-35304-3>) (see their main manuscript but also supplementary material). This is highly relevant past work that to which results should be (briefly) compared in this manuscript, as well as the conclusions made by those authors (supposedly “light” Venus lithosphere).

Response: Chen et al., 2022 assume the existence of a liquid ocean on early Venus and therefore use completely hydrated crust in contrast to the dry approach of the present-day crust in our manuscript. Due to the significant difference in the two models resulting in completely different mineral assemblages and melting temperatures, a comparison would a) have to include a more detailed discussion about water on early Venus, which we think is not useful for work we are presenting and b) would merely result in us stating that densities and melting temperatures are different due to different conditions.

- Average densities of the basaltic crust under different temperature gradients: The high crustal density values are interesting as they indeed contrast to values commonly used in geophysical/geodynamic models. I have a few minor concerns related to this paragraph:
 - Line 146 should be revised, as it references "geophysical and geodynamic models," yet only cites geophysical studies for crustal thickness values (#34-36) without including any geodynamic modeling studies that account for crustal compositions.
 - Indeed, some models like Adams et al. (2022, https://agupubs.onlinelibrary.wiley.com/doi/epdf/10.1029/2022JE007460?saml_referrer) use 2900 kg/m³ as Venus crustal density; although a set of other tectonic models (Gülcher et al. (2020; 2023) and Schools and Smrekar (2024 <https://www.sciencedirect.com/science/article/pii/S0012821X24000761>)) use equations-of-state that lead to crustal density values around 2900-2970 kg/m³, using the following equation of state: $\rho = \rho_0 [1 - \alpha(T - 298)] [1 + \beta(P - 0.1)]$ ($\rho = \rho_0 [1 - \alpha(T - 298)] [1 + \beta(P - 0.1)]$), (T in Kelvin; P in MPa) with thermal expansion $\alpha = 3 \times 10^{-5} \text{ K}^{-1}$, compressibility $\beta = 10^{-5} \text{ MPa}^{-1}$ and $\rho_0 = 3000 \text{ kg/m}^3$ as the rock density at standard temperature and pressure. Another

recent tectonic modeling work by Regorda et al. (2023 <https://agupubs.onlinelibrary.wiley.com/doi/abs/10.1029/2022JE007588>) Therefore, the values mentioned in the text (2800-2900 kg/m³) are not exactly covering the range of geodynamic models.

Response: Thank you for pointing this out. We have added the references and modified the sentence (lines 135-137) to represent the range of densities in geodynamic models and how they match our results.

• **Methods:**

o this seems very obvious, but I see nowhere stated what the surface temperature used is for the thermal profiles with the gradients? I assume 460°C based on line 26? It would be good to state in Methods regardless.

Response: Yes, that was an omission on our part. We started with surface temperature of 450 °C, which has now been added (line 325).

o Lines 350: I am under the impression the gravity value for Venus is 8.87 m/s², and not 8.77 (<https://nssdc.gsfc.nasa.gov/planetary/factsheet/venusfact.html>). Did you really use 8.77 and if so, how is this motivated? I am of the opinion that the calculations should be done with 8.87 m/s² for accuracy.

Response: Thank you for pointing this out. That was a typo, which is now corrected (line 328).